# Adolescent Athletes at Risk of Exercise-Induced Bronchoconstriction: A Result of Training or Pre-Existing Asthma?

**DOI:** 10.3390/ijerph19159119

**Published:** 2022-07-26

**Authors:** Kamila Malewska-Kaczmarek, Katarzyna Bobeff, Tymoteusz Mańkowski, Daniela Podlecka, Joanna Jerzyńska, Iwona Stelmach

**Affiliations:** 1Korczak Pediatric Center, Department of Pediatrics and Allergology, Medical University of Lodz, al. Pilsudskiego 71, 92-328 Lodz, Poland or kamila.malewska-kaczmarek@umed.lodz.pl (K.M.-K.); daniela.podlecka@umed.lodz.pl (D.P.); 2Department of Pediatrics, Oncology, Hematology and Diabetology, Medical University of Lodz, ul. Sporna 36/50, 91-738 Lodz, Poland; katarzynabobeff@gmail.com; 3Department of Radiology, Nicolaus Copernicus Regional Multi-Specialty, Oncology and Trauma Centre in Lodz, 93-513 Lodz, Poland; tymmek@gmail.com; 4Poddębice Health Care Centre, 99-200 Poddębice, Poland; w.stel@wp.pl

**Keywords:** EIB, adolescent athletes, sport

## Abstract

Exercise may trigger bronchoconstriction, especially in a group of athletes in whom bronchospasm during exercise is reported to occur more frequently than in nonathletes. The aim of this study was to determine the prevalence and environmental risk factors contributing to exercise-induced bronchoconstriction (EIB) in adolescent athletes. A prospective study was conducted among a group of 101 adolescent athletes who underwent spirometry, exercise challenge, fractional exhaled nitric oxide (FeNO) measurements, and allergy assessment. The study group was divided into three subgroups of athletes based on the most common sports environments: swimmers, “indoor” athletes, and “outdoor” athletes. The clinical evaluation demonstrated a high frequency of EIB in the study group. Moreover, a large proportion of the athletes in whom EIB was observed reported no pre-existing symptoms suggestive of bronchospasm or asthma. Among patients without a previous diagnosis of asthma, clinical evaluation confirmed 22% with positive exercise challenges, compared with 77% of adolescents with negative test results. Moreover, among the athletes with a history of asthma, 39% had positive exercise challenges. Both EIB and asthma are common conditions that affect adolescent athletes. Physicians should pay particular attention to this group, as the symptoms can lead to under- and overdiagnosis.

## 1. Introduction

It is widely known that physical exercise has a beneficial effect on children’s development. Furthermore, a lack of physical activity can lead to asthma development, particularly in children [1]. However, some adolescents choose to practice sports on an elite level. Several publications have demonstrated that the training time (in hours per week) of adolescent athletes should not exceed their age in years, e.g., a 14-year-old athlete should not train more than 14 h per week [2]. On the other hand, exercise may trigger bronchoconstriction. Exercise-induced bronchospasm (EIB) is defined as the transient and reversible narrowing of bronchi, which occurs during or after exercise. The main symptoms include dyspnea, cough, wheezing, and chest tightness [3]. Bronchospasm manifests itself as a hyperreactivity of the respiratory tract to exercise stimuli [4]. This phenomenon can be observed in both healthy and asthmatic patients. The main difference is that an asthma attack can appear during other activities, while EIB occurs only during or after exercise [3,5]. A particular group of people at risk of EIB are elite athletes, in whom bronchospasm during exercise is reported to occur more frequently than in nonathletes. It occurs due to inhalation of air pollutants and poorly conditioned air during exercise.

The prevalence of EIB in adult athletes varies between 30% and 70% [5]. Among the underaged group, the results differ: According to de Aguiar et al., the prevalence of EIB in athletes aged 5–18 years was 15%; Jonckheere et al. observed levels in athletes aged 12–13 years old at approximately 24.5% [6,7]. Goosens et al. noted that the development and frequency of EIB may be influenced by the number of years devoted to physical training [5]. In the light of recent reports, high-intensity activity may have a significant impact on the development of EIB.

Several theories describe the development of bronchoconstriction during exercise. These include thermal, osmotic, neural, epithelial damage, and inflammatory theories. When analyzing the first two mechanisms, it is believed that, during exercise, there is an increase in evaporation and a “drying” of the airway mucosa. This leads to an ionic disturbance of the mucosal barrier and the release of inflammatory mediators from nearby epithelial cells, which subsequently lyse, causing a potentially strong bronchoconstrictive effect. Upon the termination of exercise activity, the subsequent warming of the respiratory tract leads to secondary hyperemia, increased mucus secretion, and the swelling of the mucous membranes. Parasympathetic fibers, which mainly innervate the respiratory tract, according to Gawlik et al., may also be stimulated by high-intensity exercise which contributes to increased muscle tone and the subsequent EIB development. Epithelial damage also affects the development of EIB, and this is often observed in athletes. Repeated hyperventilation, combined with the inhalation of harmful pollutants from the environment, has a significant detrimental impact on the airway epithelium. This can be especially observed in swimmers who are exposed to chlorine compounds or in endurance winter sports athletes exposed to cold air [4,8].

EIB is influenced by environmental factors, such as air temperature and humidity. Children living in urban environments are 1.6 times more likely to experience EIB compared with those living in rural areas [9]. The higher rates observed in urban areas are partially explained by an increased family history of asthma symptoms or increased exposure to environmental factors in urban areas (e.g., vehicle exhaust, crowdedness, and household animals). Access to effective diagnostic tools is also limited. In addition, there is a risk that physicians will misdiagnose EIB as asthma, and subsequently, over- or undertreat the disease. Since EIB can restrict a patient’s ability to exercise and can negatively impact their quality of life, there is a growing consensus that the management of EIB needs to improve. The aim of this study was to determine the prevalence and environmental risk factors contributing to EIB in adolescent athletes in rural areas.

## 2. Materials and Methods

This is a prospective study, conducted at the Department of Pediatrics and Allergology, at the Korczak Pediatric Center in Lodz, Poland, from March 2019 to January 2022. The study included a group of Polish Caucasian children who were athletes in sports schools and clubs located in the Lodz metropolitan area. Adolescents enrolled in the research were recruited to the study by a physician during a visit to a sports facility. Some of the participants were existing patients of our outpatient clinic. The following sports were eligible for the study: football, horse riding, tennis, dance, athletics, cycling, martial arts, gymnastics, floorball, basketball, volleyball, handball, and swimming. Enrolled patients trained a minimum of four times per week for approximately 90 min per day. Exclusion criteria included: diagnosis of chronic respiratory system disease other than asthma, lack of cooperation during the lung tests, and the presence of contraindications to performing the planned tests (set according to the Polish Society of Allergology). The study was approved by the Medical Ethical Committee of the Medical University of Lodz (No. RNN/303/17/KE).

Each participant partook in two study visits. During the first visit, with written consent from parents/legal guardians and children, patients were qualified for the study. Participants were interviewed and examined by an on-site allergist. Caregivers filled a questionnaire requesting demographic data and past medical history. Children who received treatment for asthma were asked to withhold long-acting beta-2 mimetics (LABA) for a period of 24 h before the next visit. During the second visit, each patient received a skin prick test (or serum-specific immunoglobulin E (IgE) test), spirometry, pulmonary resistance test, a measurement of fractional exhaled nitric oxide (FeNO) concentration, and a standardized exercise challenge.

### 2.1. Spirometry

All pulmonary function tests were performed using a Master Screen unit (Erich Jaeger GmBH, Hochberg, Germany), as described elsewhere, in accordance with the American Thoracic Society and European Respiratory Society (ATS/ERS) guidelines [10]. During the respiratory function tests, the patients performed three forced expirations preceded by maximal inspirations. The highest of the three forced expiratory volume (FEV1) values were taken.

### 2.2. Exercise Challenge

An electronically controlled treadmill (Kettler, Ense, Germany) was used for exercise tests on all athletes. During the challenge, the following parameters were monitored: heart rate, test room temperature (a requirement was set at less than 25 °C), and room humidity (less than 50%). Heart rate monitoring was performed by a pulse meter device, a built-in feature of the treadmill.

The treadmill test lasted for 8 min with the incline of the treadmill set at 3°. The test consisted of a two-minute burn-up exercise (gradual increase in treadmill speed and participant exertion) until stabilization of 95% of the calculated maximum heart rate was achieved. This speed was maintained for another 6 min, at which point the level of exercise was decreased. At 20 and 5 min before exercise, and at 1, 3, 6, 10, 15, and 20 min after exercise, a spirometry procedure identical to the aforementioned spirometry test was performed. The analyzed parameter was the percentage of FEV1. The exercise test was performed according to the recommendations for provocative tests in allergy by the ATS/ERS. The diagnosis of EIB was made on the basis of an FEV1 decrease from the baseline of equal to or more than 10% [11,12]. All the patients with a decrease of greater or equal to 10% in FEV1% and those with symptoms during the exercise challenge (such as cough, wheezing, and chest tightness) were considered to have EIB. The results were described as ΔFEV1. This parameter provides the difference between the highest and lowest FEV1 achieved before and after the exercise challenge.

### 2.3. Fractional Exhaled Nitric Oxide Measurements

The fractional exhaled nitric oxide (FeNO) measurements were performed according to the ATS/ERS recommendations, with a chemiluminescence analyzer (model 280i fractional exhaled nitric oxide analyzer: Sievers, Boulder, CO, USA) and defined in parts per billion (ppb) [13]. The analyzer provides online, continuous measurements of FeNO per a single exhalation, with a detection range of 0.1–500 ppb. Environmental FeNO was measured before and after each test, remaining below 5 ppb. All participants were tested in a sitting position, without wearing a nose clip. Patients were asked to exhale at a constant flow rate (50 mL/s) from total lung capacity to residual volume without breath holding. A constant, expiratory flow pressure (17 cm H_2_O) was maintained by monitoring a visual display in order to eliminate contamination from nasal FeNO. Dead space and nasal FeNO (which are reflected by the FeNO concentration peak during exhalation) and FeNO from the lower respiratory tract (determined by the plateau value after the peak) were automatically recorded by using the manufacturer’s software. Three FeNO measurements of the plateau phase were obtained, with less than 10% variation. The mean value of three successive, reproducible recordings was retained for statistical analysis. The recommendations used for the analysis were based on the study by Buchvold et al. [14].

### 2.4. Diagnosis of Atopy

Skin prick tests with airborne allergens (using Allergopharma, Reinbeck, Germany) were performed on all patients. Alternatively, blood tests (IgE antibody test using Polycheck Allergy (Biocheck GmbH, Münster, Germany)) against airborne allergens were assessed. The material was obtained after sampling 10 mL of peripheral blood. The type of test depended on patients’ cooperation.

### 2.5. Follow-Up

At the end of the tests, each athlete underwent a physical examination. Two doses of salbutamol were administered if the patient exhibited cough, wheezing, or shortness of breath. Children who required further medical diagnosis and treatment were admitted to the hospital.

### 2.6. Statistical Analysis

Statistical analyses were performed using Statistica software (Version 13; StatSoft, Inc., Tulsa, OK, USA). The distribution of the data was assessed with the Shapiro–Wilk test, and *p* > 0.1 was considered to indicate normal distribution. The median instead of the mean was used for comparison in all cases in regard to distribution other than normal. Nonparametric tests were used for the skewed distributions. U Mann–Whitney and chi-squared tests with or without the Yates correction and the two-tailed exact Fischer’s test were applied to determine the differences between the two groups according to the type of data. A comparison between multiple groups was performed with a nonparametric analysis of variance (Kruskal–Wallis ANOVA); if ANOVA yielded a significant difference, this was followed by between-group multiple comparisons with a nonparametric post hoc test. For all tests, *p* < 0.05 was deemed significant, although we also present results with 0.05 < *p* < 0.1 in the Kruskal–Wallis nonparametric analysis section.

## 3. Results

In this study, 39 girls and 62 boys (101 in total) adolescent athletes, aged 12–18 years, were included. There were no statistically significant results concerning EIB between the female and male gender. All the details of patient characteristics are shown in Table 1.

When dividing patients into groups, namely athletes performing “outdoor” activities, those performing “indoor” activities, and swimmers (Table 2), symptoms during or after exercise were observed in 54 patients; specifically, dyspnea was noted in 35 patients: outdoor athletes—21% of patients, indoor athletes—47%, and swimmers—38% (*p* = 0.04815). Moreover, in these groups, 21% of athletes from outdoor activities, 42% from indoor activities, and 14% of swimmers (*p* = 0.03605) had positive and very close-to-positive results for EIB. Prior to the study, 7 soccer players (18%) observed dyspnea during or after exercise, in comparison with 28 other patients (45%) (*p* = 0.00979). All the details concerning this division are included in Table 2 and Figure 1.

### 3.1. Diagnosis of Asthma and EIB

Asthma was diagnosed in 24 athletes. While analyzing individual sports, only four soccer players (10%) had a previous history of asthma (*p* = 0.01548). A diagnosis of EIB was confirmed in 28% (*n* = 28) of adolescent athletes: 61% without asthma and 39% with asthma. All the data concerning asthma, symptoms, the diagnosis of EIB, and treatment are summarized in Table 3 and Figure 2.

### 3.2. Lung Tests

#### 3.2.1. Spirometry Characteristics

During the spirometry analysis, significant differences were observed among examined groups: Swimmers achieved the highest FEV1 values, followed by indoor athletes, whose results were lower but still within a normal range (*p* = 0.0295). Furthermore, a similar pattern was observed for forced vital capacity (FVC) (*p* = 0.0257) (Figure 3). Pulmonary function tests and changes among groups are presented in Table 1 and Table 4.

#### 3.2.2. FeNO Characteristics

Among all the examined athletes, 89 (88%) had significantly elevated fractional exhaled nitric oxide values. In this group, we did not detect the presence of atopy in 40 (45%) adolescents (*p* = 0.0588). All the details concerning FeNO in athletes are included in Figure 4.

#### 3.2.3. Atopy

During the analysis of atopy in the patients, higher ΔFEV1 values were achieved by athletes diagnosed with atopy to grass (*p* = 0.004021), trees (0.017704) (measurements were performed outside of the pollen season), and mold allergens (0.007743). In patients with a history of asthma, 4 athletes (10%) were found to have no atopy, compared with 20 athletes (33%) who were allergic to at least one allergen (*p* = 0.00898) and 17 (37%) with polyvalent atopy (0.00894). A similar analysis was performed for patients with positive exercise challenges without asthma: The results revealed that 12 athletes (30%) had no atopy, and 16 patients (26%) were allergic to at least one allergen; however, these results were not statistically significant (*p* = 0.85185).

## 4. Discussion

This study highlights the high prevalence of EIB in adolescent athletes. In the athletes from the Lodz metropolitan area, the prevalence was 28%. Various factors influence the pathogenesis of exercise-induced bronchoconstriction. This was particularly evident in the 28 patients with an EIB diagnosis, where as many as 17 did not display asthma symptoms. A question then arises: Why do adolescents who train athletically often have EIB symptoms but are not diagnosed with asthma? Is it undiagnosed asthma or are the environmental conditions stimuli for the EIB development? Aggarwal et al. emphasized that the urban environment can have an impact on EIB occurrence in children [3]. This has also been observed in our athletic population, for instance, in the athletes from the indoor group. Adolescents who trained indoors (sports hall) were interestingly more likely to be diagnosed with EIB. These results may be due to higher air pollution in the Lodz agglomeration in Poland, where our patients trained. These conditions certainly contribute to environmental pollution and health consequences in adolescents [15]. In a previous research study from our Department, we determined strong predictors for EIB. That research was conducted among children in a school environment. We observed that elevated barometric pressure and higher humidity levels exerted strong influences on the occurrence of EIB in nonathletic schoolchildren. Furthermore, the study revealed that indoor cat dander allergens, discovered on precipitants’ clothes within the school and the sports hall, increase the risk of bronchoconstriction. Moreover, several studies indicate that asthma symptoms may be related to mold exposure, which is mostly found indoors. Our findings highlight that athletes allergic to mold, grass, and tree pollen are more prone to EIB than athletes without such allergies. These findings may explain why 47% of indoor athletes are EIB-positive and have worse results compared with athletes of other sports groups [16].

The above results underline the importance of the environment in which young athletes exercise. Rapid and deep breathing, adverse allergenic factors (e.g., mold, animal allergens), and nonallergenic factors (e.g., pollution, dry air) may affect the epithelium of the respiratory tract, potentially leading to the formation of EIB. The results of 40 nonatopic athletes in whom FeNO was elevated support the above statement. On this basis, we argue that nonallergic factors can also induce inflammation, including the well-known biomarker FeNO. This contradicts a large amount of scientific research, according to which higher FeNO levels are related to atopy. However, it is worth noting that FeNO levels are affected by other factors, such as age, sex, height, current smoking status, and diet [17].

### 4.1. Type of Training

Although exercise is indicated in the prevention of asthma, it is emphasized that excessive airway stress can damage the airway epithelium, leading to chronic inflammation and bronchoconstriction [5]. Particularly in athletes without asthma, the epithelial injury theory is hypothesized to be a contributing factor in the EIB development, due to the chronic inhalation of air pollutants or chlorine products (swimmers). Furthermore, due to intense air ventilation, the osmotic theory is believed to contribute to bronchoconstriction. Moreover, cold air can cause the dehydration of bronchi, which is also related to EIB development [3,5,8]. These examples support our claim that different types of sports involve different degrees of EIB development.

Our study revealed that indoor athletes achieved lower spirometry parameters compared with other athletes. Moreover, athletes practicing outdoor activities achieved better results; within this group, 21% had positive EIB results in comparison to 42% of indoor athletes. Additionally, a history of dyspnea during training was reported by 21% of outdoor athletes, compared with 47% of indoor athletes and 38% of swimmers.

According to Ventura et al., the prevalence of asthma in soccer players was lower than the prevalence in other elite athletes [18]. Their study identified a similar relationship: Only 10% of soccer players had a history of asthma in comparison to other athletes, in whom results were higher. Furthermore, only 18% of soccer players had a history of dyspnea during exercise. In our division, football players were categorized in the outdoor-activity group, because most of the time, they trained outside. Therefore, these results raise several questions: Are football players a group with a lower incidence of asthma? Or, is this because they are less likely to report symptoms due to the social pressure and popularity of the sport? Due to the popularity of soccer, we wish to focus on these questions in future research.

Ventura et al. suggested that the type of sport may influence the incidence of asthma, which has so far been observed more frequently in endurance athletes, e.g., long-distance runners or swimmers [18]. It is clearly indicated here that the conditions in which athletes trained had a major impact on the EIB development.

### 4.2. Under- and Over-Diagnosis of Bronchoconstriction

Some authors note the benefits of athletes completing questionnaires to help diagnose EIB. This approach comes with certain limitations, as athletes occasionally confuse EIB symptoms with the body’s normal response to exercise [5]. Burnett et al. argued that the majority of athletes report no EIB symptoms [19]. In our research, in 28 patients with confirmed EIB in exercise challenges, 7 (25%) denied exercise-induced symptoms. This may be due to a lack of awareness or ability to perceive EIB symptoms [5].

On the other hand, our study revealed that 45% of athletes who reported symptoms of EIB had a negative exercise challenge, among which 15% observed cough prior to study inclusion, and 27% reported dyspnea during or after exercise. As previous studies indicate, there is a need to consider alternative causes of EIB symptoms, such as exhaustion, exercise-induced laryngeal dysfunction, and exercise-induced hyperventilation [5]. Consequently, many physicians may misdiagnose EIB symptoms as asthma, contributing to under- and overtreatment [3]. This was particularly evident in the results of our study: Within the group of athletes who received antiasthmatic treatment (ICS-LABA), only 15% had a negative exercise test result, which may have been due to proper disease control. Some authors argue that using self-reported symptoms in EIB diagnosis and treatment may lead to an inaccurate diagnosis [5,20].

### 4.3. EIB as a Result of Training or Already an Asthma?

The observation of the EIB phenomenon in athletes raises the question of who may develop asthma. The typical symptoms of EIB are wheezing, shortness of breath, cough, and tightness in the chest. Patients with asthma also experience such symptoms, raising the question of whether to consider an isolated case of EIB or asthma. According to the definition by the Joint Task Force on Practice Parameters in Allergy and Immunology, EIB is related to transient bronchoconstriction during or after exercise. EIB may or may not be related to asthma attacks, a relation previously termed exercise-induced asthma (EIA) [21]. For this reason, the division into EIA and EIB is no longer considered in recent publications. An alternative classification into EIB with asthma (EIBa) and EIB without asthma (EIBwa) has been proposed [5].

According to Aggarwal et al., the prevalence of EIB in patients with asthma is approximately 90% [3]. Our patients with a history of asthma showed a 39% frequency of EIB, which may be associated with the development of the disease. On the other hand, 22% of athletes with a positive exercise challenge had never been diagnosed with asthma. This raises another question: Is undiagnosed asthma or another pathology causing bronchospasm, given that 75% of these patients had a history of EIB symptoms? In our study, 39% of the patients with confirmed EIB had a pre-existing cough during exercise, and within that group, dyspnea was reported by 54%. Some authors note that using self-reported symptoms in the diagnosis of EIB leads to an inaccurate diagnosis [5,20]. In such cases, another question should be raised: How does one detect such patients? Should a reversibility test be performed to confirm asthma in every case of an athlete with a positive exercise challenge?

### 4.4. Measurements of Fractional Exhaled Nitric Oxide

FeNO measurement is considered to be an effective, noninvasive tool to assess eosinophilic airway inflammation. In our observations, this parameter reached the lowest values among outdoor athletes, compared with higher values in athletes training indoor activities. Among the examined adolescents, FeNO values for swimmers had intermediate results. It is important to remember that the pattern of inflammation in swimmers is neutrophilic; consequently, the role of FeNO in this group is ambiguous [5].

In our research, FeNO levels in athletes with confirmed EIB were higher than those in adolescents in whom EIB was not yet confirmed. This result was not statistically significant, with a resulting smaller sample of respondents than initially assumed. According to Errson et al., among the group of EIB-confirmed adolescent athletes, FeNO was not related to the disease. This leads us to consider that the mechanisms of EIB in athletes and the general population are different [21].

### 4.5. Allergic Status

Our study revealed that the greater the decline in FEV1 during an exercise test in adolescent athletes, the greater the levels of atopy to grass, trees, and mold. According to Goossens et al., among young athletes, the prevalence of atopy is elevated [5]. In a Tunisian study, researchers noted that atopy was a major risk factor in the diagnosis of EIB [22]. Our study revealed a high prevalence of atopy (33%) and polyvalent atopy (37%) among athletes with previously diagnosed asthma. Similarly, when analyzing the results of patients with a positive exercise challenge, atopy was discovered in 26% and polyvalent in 33% of adolescents; however, these results were not significant. We speculate that in our study group, allergens did not exhibit significant effects on the development of EIB. If airborne allergens did indeed have an effect, it was only in atopy to grass, trees, and mold.

Based on the analysis of our collected data, we argue that, during exercise, factors such as inhaling cold air are the trigger for bronchospasm rather than airborne allergens. In the case of year-round mold allergens, one must consider whether these are responsible for inferior test results in indoor athletes.

### 4.6. Limitations and Study’s Strengths

During our study, no reversibility test was performed to confirm asthma; this will be the subject of a future study. The less-than-optimal sample size of 101 patients was due to limitations incurred during the SARS-CoV2 pandemic. Another limitation was that patients on permanent medication were not reassessed for whether these existing treatments were indeed appropriate. Furthermore, the authors intended to identify environmental risk factors that promote EIB, but due to the pandemic, such opportunities were greatly restricted. Nonetheless, our study confidently analyzed the results from athletes in different disciplines and environments. We evaluated several factors that potentially contribute to the development of EIB. Our efforts seek to help physicians more properly diagnose suspected patients with EIB.

## 5. Conclusions

Our research provided significant evidence that EIB is a common phenomenon among adolescent athletes. Our first “take-home” message is that not every athlete may report or be aware of symptoms of bronchospasm. This may lead to underdiagnosis and result in improper treatment. Secondly, we wish to draw attention to unsuitable conditions for adolescent athletes during training and highlight typical symptoms, such as cough and dyspnea. These factors may indicate EIB development. Our study concludes with two open questions: Could EIB-positive athletes develop asthma in the future? What treatment strategies should be employed in such patients?

## Figures and Tables

**Figure 1 ijerph-19-09119-f001:**
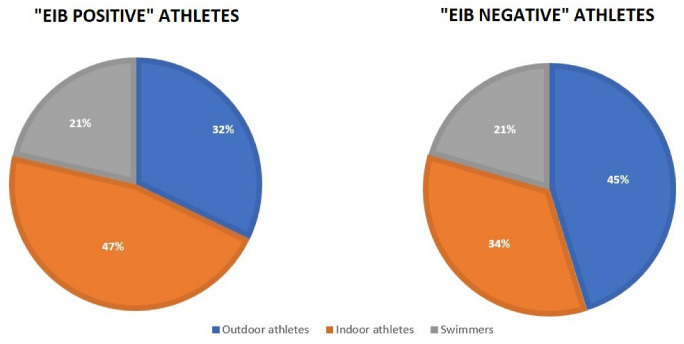
Differences between study groups among EIB-positive and EIB-negative patients.

**Figure 2 ijerph-19-09119-f002:**
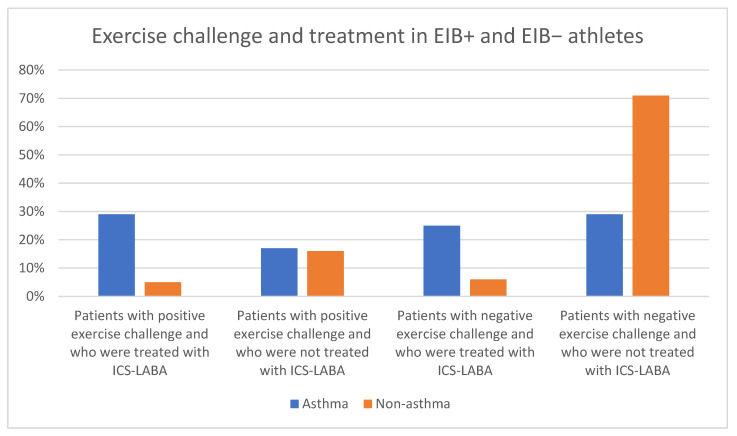
Exercise challenge and treatment in EIB+ and EIB− athletes.

**Figure 3 ijerph-19-09119-f003:**
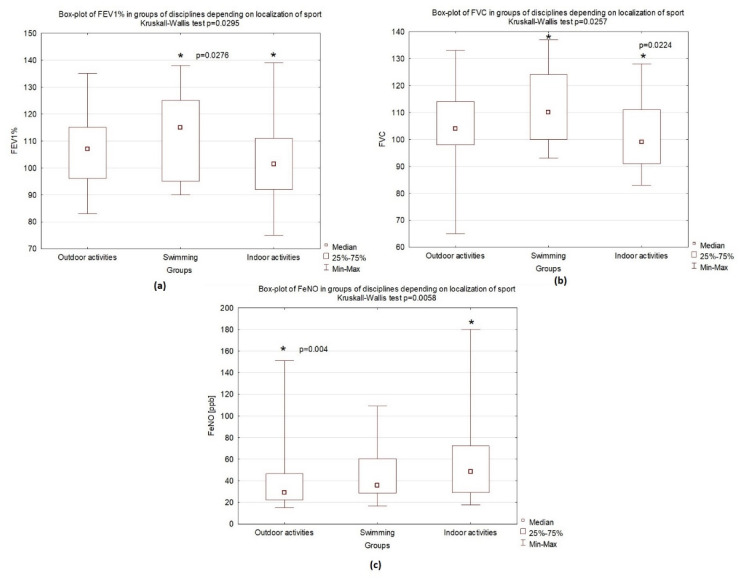
Differences between study groups: median, interquartile range (25–75%), and minimum and maximum values: (**a**) box plot of FEV1 in groups of disciplines depending on the sports environment; (**b**) box plot of FVC in groups of disciplines depending on sports environment; (**c**) box plot of FeNO depending on sports environment. * statistically significant result.

**Figure 4 ijerph-19-09119-f004:**
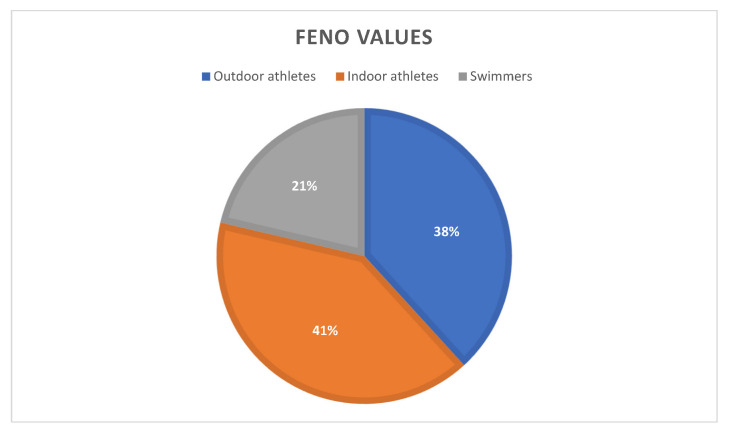
Athletes with increased FeNO divided by place of training of study groups.

**Table 1 ijerph-19-09119-t001:** Patient characteristics.

Variables	
Age (years), mean ± SD	13.9 ± 1.7
Female gender, *n*(%)	39 (38.6)
Male gender, *n*(%)	62 (61.3)
Body weight (kg), mean ± SD	54.9 ± 14.4
Height (cm), mean ± SD	163.6 ± 16
BMI (kg/m^2^) *, mean ± SD	20.1± 2.7
FEV_1_ (% best/pred.) **, mean ± SD	106.1 ± 14.4
FVC (% best/pred.) ***, mean ± SD	105.3 ± 13.6
FeNO (ppb) ****, mean ± SD	49.3 ± 36

* Body mass index; ** forced expiratory volume in 1 s; *** forced vital capacity; **** fractional exhaled nitric oxide.

**Table 2 ijerph-19-09119-t002:** Distribution of athletes in the study depending on their practiced discipline.

Variables		
Outdoor athletes	Football, *n*(%)	39 (39)
	Horse riding, *n*(%)	3 (3)
Indoor athletes	Tennis, *n*(%)	2 (2)
	Dance, *n*(%)	4 (4)
	Athletics, *n*(%)	4 (4)
	Cycling, *n*(%)	4 (4)
	Martial arts, *n*(%)	8 (7.9)
	Gymnastics, *n*(%)	1 (1)
	Floorball, *n*(%)	1 (1)
	Basketball, *n*(%)	10 (9.9)
	Volleyball, *n*(%)	2 (2)
	Handball, *n*(%)	2 (2)
Swimmers	Swimming, *n*(%)	21 (21)

**Table 3 ijerph-19-09119-t003:** Details concerning EIB-positive and EIB-negative athletes.

	EIB (+) (*n* = 28)	EIB (−) (*n* = 73)	*p*
**Asthma** (*n* = 24)	11 (39%)	13 (18%)	0.04455
**Nonasthma** (*n* = 77)	17 (61%)	60 (82%)	
**Pre-existing symptoms** (*n* = 54)	21 (75%)	33 (45%)	0.00721
**Athletes without pre-existing symptoms** (*n* = 47)	7 (25%)	40 (55%)	
**Athletes reporting cough during exercise** (*n* = 22)	11 (39%)	11 (15%)	0.01778
**Athletes reporting dyspnea during exercise** (*n* = 35)	15 (54%)	20 (27%)	0.02504
**Athletes who take ICS-LABA *** (*n* = 22)	11 (39%)	11 (15%)	0.01778

* Inhaled corticosteroids–long-acting beta-agonists.

**Table 4 ijerph-19-09119-t004:** Spirometry parameters by athlete groups.

	Outdoor Athletes			Swimmers			Indoor Athletes			
	Median	25%	75%	Median	25%	75%	Median	25%	75%	*p*
FEV1 %	107	96	115	115	95	125	101.5	92	111	*p* = 0.0295
FEV1/FVC % *	100	91	106	102	96	107	100.5	96	106	*p* = 0.5898
PEF % **	95	86	111	98	92	104	89	81	99	*p* = 0.0486
FVC	104	98	114	110	100	124	99	91	111	*p* = 0.0257
MEF25 ***	84	68	94	93	74	115	87	65	103	*p* = 0.4659
MEF50 ***	93	78	110	104	73	115	85	70	103	*p* = 0.2356
MEF75 ***	100	86	111	99	89	116	94	81	102	*p* = 0.1025
FeNO	29	22.3	46.5	35.8	28.6	60.4	48.5	29.3	72.3	*p* = 0.0058
Resistance %	144.5	127	161.5	144	114	180	145.5	131	166	*p* = 0.8741

* Forced expiratory volume in one second % of vital capacity; ** peak expiratory flow; *** maximal expiratory flows at 25%, 50%, and 75% of FVC.

## Data Availability

Not applicable.

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
