# Peer review of "Adolescent Athletes at Risk of Exercise-Induced Bronchoconstriction: A Result of Training or Pre-Existing Asthma?"

_ijerph, 2022, doi:10.3390/ijerph19159119_

Round 1

Reviewer 1 Report

Adolescent athletes at risk of exercise induced bronchoconstriction result of training or already asthma?

Abstract – good

Introduction with aim – OK "EIB is a common phenomenon among adolescent athletes" – it it already known, it would be better to have more than 101 adoloscent.

Probably, due to Covid19 pandemia it would be better results, more than 101

Method/materials OK

Results – OK

Discussion –OK

Literature- 2022-2017-11/22 - excellent

2016-2011-5/22

2011-oldier-5/22

The topic of the paper is very well known and elaborated.
The result must contain a much larger number of patients from different sports.
The paper can be published without major corrections

Author Response

Dear reviewer, 

Thank you for your valuable comments. 

Kind regards

Reviewer 2 Report

The authors aimed to determine the prevalence and environmental risk factors contributing to exercise-induced bronchoconstriction (EIB) in adolescent athletes in the rural area.

The study covers some issues that have been overlooked in other similar topics. The structure of the manuscript appears adequate and well divided in the sections. Moreover, the study is easy to follow, but some issues should be improved. Some of the comments that would improve the overall quality of the study are:

a. Authors must pay attention to the technical terms acronyms they used in the text.

b. English language needs to be revised.

c. Conclusion Section: This paragraph required a general revision to eliminate redundant sentences and to add some "take-home message".

Author Response

Dear reviewer, 

Thank you for your valuable comments.

You have suggested paying attention to technical terms and acronyms, which we corrected. We also corrected the last part of article “conclusions” and added “take-home messages” as you suggested. English language was revised by english native speaker. 

Thank you for your time and we hope that our article fulfills requirements.

Kind regards,  

Reviewer 3 Report

The manuscript introduced bronchoconstriction induced by exercise in adolescent athletes. The authors showed symptoms by tests using an electronically controlled treadmill, between exercise-induced bronchoconstriction-positive and negative athletes. indoor athletes were apt to be exercise-induced bronchoconstriction-positive. Thus, these findings will be useful for physical exercise in adolescent athletes. Therefore, the manuscript is not too excellent to be published. In other words, the manuscript is so excellent that it should be published.

Comments

(1) How much amount of physical exercise was suitable in adolescent athletes?

(2) In what mechanisms did physical exercise cause bronchoconstriction?

(3) Were there any racial differences in exercise-induced bronchoconstriction?

(4) Were there any gender differences in exercise-induced bronchoconstriction?

(5) Did an electronically controlled treadmill cast stress on exercising adolescent athletes?

(6) Why did exercise-induced bronchoconstriction-positive athletes largely exist in indoor athletes, compared to negative ones? Indoor cat dander allergens?

That is all.

Author Response

Dear reviewer, 

Thank you for your valuable comments.

  1. “How much amount of physical exercise was suitable in adolescent athletes?” we added this information in “Introduction”.
  2. “In what mechanisms did physical exercise cause bronchoconstriction?” – precise mechanisms were described in Introduction: “Several theories describe the development of bronchoconstriction during exercise. These are: thermal, osmotic, neural, epithelial damage and inflammatory theories …”. What is more, we described mechanisms in "Discussion": "Type of training"
  3. “Were there any racial differences in exercise-induced bronchoconstriction?” - as the Polish population is homogeneous, there were no racial differences between the athletes in this research. We added information that all our patients are Caucasian in “Materials and methods”
  4. “Were there any gender differences in exercise-induced bronchoconstriction?” - these results were analyzed but they were not statistically significant. We added this information in the first paragraph of “results”
  5. “Did an electronically controlled treadmill cast stress on exercising adolescent athletes?” - the detailed impact of the treadmill test on athletes is described in “Materials and methods”: “The treadmill test lasted for 8 minutes with the incline of the treadmill set at 3º. The test consisted of two minute burn up exercise (gradual increase in treadmill speed and participant exertion) until stabilization of 95% of the calculated maximum heart rate was achieved. This speed was maintained for another 6 minutes, at which point the level of exercise was decreased.”
  6. “Why did exercise-induced bronchoconstriction-positive athletes largely exist in indoor athletes, compared to negative ones? Indoor cat dander allergens?” – it was precisely described in the first paragraph in “Discussion”. We made some changes to make it clear.

English language was revised by english native speaker. 

Thank you for your time and we hope that our article fulfills requirements.

Kind regards,  

Reviewer 4 Report

The theme of this study is very interesting because today much importance is given to physical exercise for his beneficial effect on children's development.  

The aim of the study is well described in the Abstract. Authors want to determine the prevalence and environmental risk factors contributing to exercise-induced bronchocostriction in adolescent athletes, and they fully describe all the possible pathogenetic mechanisms underlying bronchospasm. The introduction clarified that bronchospasm may be expressed in hyperreactivity of respiratpry tract due to the stimulus of exercise, and that this phenomenon can be observed both in healthy and asthmatic patients. the difference is that asthma attacks appear during other activities, while exercise-induced bronchospasm occurs only during or after exercise.

Inclusion criteria in the study are precise: all elegible sports are listed and the minimun weekly frequency is also specified. the ways in which the subjects were studied in the two scheduled visits are also clearly explained.

The authors very critically also question the possible overdiagnosis of bronchocostriction and they consider also some alternative causes of EIB symptoms, such as axahaustion or exercise hyperventilation.

The study perhaps could have included a greater number of included adolescents, especially because it is important to understand if bronchospasm is the result of physical exercise or if is related whit a not diagnosed asthma. 

in conclusion, the study emphasized the importance of following subjects with EIB Symptoms over time in order to highlight and therefore make an early diagnosys.

Author Response

Dear reviewer, 
Thank you for your valuable comments. Our article was revised by english native speaker. Unfortunately, due to pandemic we did not have opportunity to recruit more participants. We hope to continue our research in future.

Kind regards,

This manuscript is a resubmission of an earlier submission. The following is a list of the peer review reports and author responses from that submission.